# Semi-supervised Bayesian integration of multiple spatial proteomics datasets

**Stephen Coleman** [1]*, **Lisa Breckels**[2], **Ross F. Waller**[2], **Kathryn S. Lilley**[2], **Chris Wallace**[1,3,4], **Oliver M. Crook**[5,6☉], **Paul D. W. Kirk**[1,3,4☉]

**1** MRC Biostatistics Unit, University of Cambridge, Cambridge, United Kingdom, **2** Department of Biochemistry, University of Cambridge, Cambridge, United Kingdom, **3** Department of Medicine, University of Cambridge, Cambridge, United Kingdom, **4** Cambridge Institute of Therapeutic Immunology and Infectious Disease, University of Cambridge, Cambridge, United Kingdom, **5** Department of Chemistry, University of Oxford, Oxford, United Kingdom, **6** Kavli Institute for Nanoscience Discovery, University of Oxford, Oxford, United Kingdom

☉ These authors contributed equally to this work.
* stephen.d.p.coleman@gmail.com

## Abstract

The subcellular localisation of proteins is a key determinant of their function. High-throughput analyses of these localisations can be performed using mass spectrometry-based spatial proteomics, which enables us to examine the localisation and relocalisation of proteins. Furthermore, complementary data sources can provide additional sources of functional or localisation information. Examples include protein annotations and other high-throughput 'omic assays. Integrating these modalities can provide new insights as well as additional confidence in results, but existing approaches for integrative analyses of spatial proteomics datasets, such as concatenation-based methods and transfer learning approaches like KNN-TL, are limited in the types of data they can integrate and do not quantify uncertainty in their predictions. Here we propose a semi-supervised Bayesian approach (wherein model parameters are inferred from *both* labeled marker proteins and unlabeled data while quantifying prediction uncertainty) to integrate spatial proteomics datasets with other data sources, to improve the inference of protein sub-cellular localisation. We demonstrate our approach outperforms other transfer-learning methods and has greater flexibility in the data it can model - including categorical annotations (e.g., Gene Ontology terms), continuous measurements (e.g., protein abundance), and temporal profiles (e.g., time-series expression data). To demonstrate the flexibility of our approach, we apply our method to integrate spatial proteomics data generated for the parasite *Toxoplasma gondii* with time-series gene expression data generated over its cell cycle. Our findings suggest that proteins linked to invasion organelles are associated with expression programs that peak at the end of the first cell-cycle.

**Data availability statement:** Project specific scripts:

https://github.com/stcolema/tagmmdi_wd. R package for main method: https://github.com/stcolema/MDIr. The datasets analyzed in this study are publicly available. The cell cycle gene expression data [Behnke et al., 2010] is available through https://www.omicsdi.org/dataset/geo/GSE19092. The spatial proteomics data are available through the pRolocdata Bioconductor package [Gatto et al., 2014]. Specifically, under the names: • T. Gondii, ToxoLopit dataset [Barylyuk et al., 2020]: https://www.cell.com/cms/10.1016/j.chom.2020.09.011/attachment/5a810623-063a-4966-8c4f-39ab8bcf2eee/mmc3.xls. • Human embryonic kidney cells (HEK293T2011 dataset) and Mouse pluripotent stem cells (E14TG2aS1 dataset) [Christoforou et al., 2012]: https://www.ebi.ac.uk/pride/archive/projects/PXD001279. • Arabidopsis thaliana root tissue (groen2014r1 dataset) [Groen et al., 2014]: https://pubs.acs.org/doi/suppl/10.1021/pr4008464/suppl_file/pr4008464_si_006.xls. • Arabidopsis thaliana callus tissue (dunkley2006 dataset) [Dunkley et al., 2006]: https://www.pnas.org/doi/suppl/10.1073/pnas.0506958103/suppl_file/06958table2.xls. • Human cytomegalovirus-infected fibroblasts at 24, 48, 72, 96, and 120 hours post-infection (beltran2016 series) [Beltran et al., 2016]: https://proteomecentral.proteomexchange.org/cgi/GetDataset?ID=PXD003925. • Human cancer cell lines A431, NCI-H322, HCC-827, MCF7, U251 (orre2019 series) [Orre et al., 2018]: https://www.ebi.ac.uk/pride/archive/projects/PXD006895.

**Funding:** This research was funded in whole, or in part, by the MRC (MC_UU_00040/01 to CW, MC_UU_00040/05 to PDWK, MR/Y010078/1 to OMC) and the Wellcome Trust (WT107881 to CW, 214298/Z/18/Z to RFW). OMC

Furthermore, this integrative analysis divides the dense granule proteins into heterogeneous populations suggestive of potentially different functions. Our method is disseminated via the `mdir` R package available on the lead author's Github.

## Author summary

Proteins are located in subcellular environments to ensure that they are near their interaction partners and occur in the correct biochemical environment to function. Where a protein is located can be determined from a number of data sources. To integrate diverse datasets together we develop an integrative Bayesian model to combine the information from several datasets in a principled manner. We learn how similar the dataset are as part of the modelling process and demonstrate the benefits of integrating mass-spectrometry based spatial proteomics data with timecourse gene-expression datasets.

## Introduction

Proteins are key to all cellular processes including cell proliferation and survival. To function correctly, a protein must interact with other binding partners and substrates which requires them to localise to the correct subcellular compartment [1]. For example, signalling and metabolic pathways are dependent on the location of the constituent proteins [2]. Mislocation to the incorrect compartment is increasingly associated with disease, including cancer and obesity [3–7]. Thus, understanding the location of proteins within the cell is of use in the development of therapeutic targets and determining disease aetiology [8]. Mass-spectrometry (**MS**) based spatial proteomics can be used to study the localisation and relocalisation of proteins on a proteome- and system-wide scale. These experiments leverage so-called marker proteins with an unambiguous single localisation and then apply machine learning approaches, such as the support vector machine (**SVM**), to predict the localisation of annotated proteins.

In a typical MS-based spatial proteomics experiment cells are first gently lysed, to dissociate the cellular compartments while preserving organelle integrity. This cellular material is then separated into subcellular fractions using one of a range of methods such as differential centrifugation [9,10] or equilibrium density centrifugation [11,12]. Following subcellular fractionation, the proteins are digested to peptides and subject to high accuracy quantitative MS generating a signature of abundance across the subcellular fractions which correlates with subcellular localisation. A wide variety of methods exploit these central principles, including protein correlation profiling, SubCellBarCode, dynamic organeller maps amongst others [9,11,13–15]. In the localisation of organelle proteins by isotope tagging (**LOPIT**) [11,16] and hyper-plexed LOPIT (**hyper-LOPIT**) [12,17] approaches, cell lysis is preceded by the separation of subcellular components along a continuous density gradient. Discrete fractions along this gradient are then collected and multiplexed quantification is achieved using tandem mass tags (**TMT**) [18]. Protein distributions revealing organelle specific

acknowledges funding from a Todd-Bird Junior Research Fellowship. LB was supported by EU Horizon 2020 programme INFRAIA project EPIC-XS (project 823839). CW is a part-time employee of GSK; GSK had no involvement in or influence on the work presented here. The funders had no role in study design, data collection and analysis, decision to publish, or preparation of the manuscript. For the purpose of Open Access, the author has applied a CC BY public copyright license to any Author Accepted Manuscript version arising from this submission.

**Competing Interest:** The authors have declared that no competing interests exist.

profiles within the fractions are acquired by using synchronous precursor selection mass-spectrometry (**SPS-MS3**). Proteins belonging to the same organelle possess shared quantitative abundance profiles [19,20], it is these (normalised) abundance profiles that are the input to machine learning algorithms to predict the localisation of the unannotated proteins. This means the marker proteins form a labelled subset of the data which can be used as a training set for supervised methods (e.g., partial least squares discriminant analysis [16,21]; the support vector machine [9,12]). Using a supervised method assumes that both the marker proteins are fully representative of their compartment, that all compartments have known marker proteins and that all subcellular niches are observed in the labelled set. Supervised classification remains the most common type of Machine Learning approach to prediction tasks in proteomics [22]. If we do not believe these assumptions hold, we can employ semi-supervised methods to both infer the unobserved localisations and update the parameters describing the components simultaneously [23,24]. These methods can typical uncover unannotated subcellular niches and have better accuracy at inferring the underlying model parameters.

'Omic technologies that infer, say, gene expression, report on different properties of proteins that can help us to understand the aetiology of diseases and conditions that cause functional changes in the system, e.g., cancers, neurodegenerative diseases and immune-mediated diseases, etc. [25–30]. Deepening understanding of gene expression might elucidate the mechanisms by which viruses and some parasites hijack host organelles to replicate [31]. Additionally they can have implication for disease diagnosis and therapeutic success [32,33]. More generally, 'omic measurements can give us an integrated view on the molecular mechanisms of biological processes [34]. However, integrating diverse datasets (sometimes referred to as modalities in the machine learning literature) is challenging because the different measurement processes generate heterogeneous data structures and each method has its own set of limitations. Current integrative approaches have significant limitations for spatial proteomics applications. Simply concatenating the datasets assumes each dataset contributes equal information towards inference and ignores dataset specific variability.

Despite the successes of multi-omic data integration [35], most methods are unsupervised and tend to infer latent factors that explain variation in the data; that is, a de-convolution of the dataset into a low-dimensional representation [36,37] or to identify shared clusters [38–40] but cannot leverage the valuable marker protein annotations available in spatial proteomics experiments. Transfer learning approaches like k-nearest neighbors [41] can incorporate auxiliary data but provide point estimates without uncertainty quantification and are limited to pairwise dataset comparisons.

Bayesian methods are particularly well-suited to address these limitations because they provide a principled framework for: (1) incorporating prior biological knowledge through marker proteins in a semi-supervised setting, (2) quantifying uncertainty in protein localisation predictions, (3) learning the degree of information sharing between datasets rather than assuming it, and (4) handling heterogeneous data types through modality-specific probability models [42]. Here, we perform integrative

analysis of MS-based spatial proteomics data with other datasets (e.g. Gene Ontology (**GO**) annotations and Gene-expression datasets) as they contain powerful signals that may uncover subcellular regulation of processes that are not apparent from considering another dataset in isolation. Furthermore, the signal in one MS-based spatial proteomics dataset may be boosted by considering multiple datasets simultaneously. More generally, we wish to jointly model structure in across datasets and uncover shared clusters, groups or regulation while properly accounting for uncertainty and dataset-specific characteristics.

When considering pairs of datasets (though we can consider larger combinations), we can have one of the following three scenarios. Either both our modalities have some observed classes, neither do, or one modality does. MS-based spatial proteomic data are annotated with marker proteins and so at least one of our modalities contains some known structure which should be incorporated into the modelling. This motivates developing semi-supervised integrative methodology and account for dataset-specific models. Our approach is applicable beyond MS-based spatial proteomics to general semi-supervised integrative tasks (such as the integration of clinical measurements with genomic data in personalized medicine applications) and is implemented in an R-package: `mdir` available from https://github.com/stcolema/MDIr.

The paper is organised as follows. In the Materials and methods section the models used and data analysed in this paper are presented. This includes a description of the simulation study comparing some of the models, the datasets used to validate our approach on real data and our case study, two 'omics datasets for the model apicomplexan, *Toxoplasma gondii*. The Results section deals with the results from analysing these data showing the advantage of leveraging information from observed markers/labels and across datasets via integrative modelling. In the analysis of *T. Gondii*, we find evidence for dividing the dense granule proteins into heterogeneous populations with potentially different functions. Finally, in Discussion we consider the results described in preceding sections.

## Materials and methods

### Methods

We introduce Gaussian mixture models as the foundation of the TAGM [23], GP mixture and (multiple dataset integration) MDI [40] models, which we then introduce in turn. Finally we introduce the KNN transfer learning algorithm, a supervised method for multi-modal data in spatial proteomics.

**Mixture models.** Let $X = (X_1, \dots, X_N)$ be the observed data, with $X_n = [X_{n,1}, \dots, X_{n,P}]^\top$ for each item being considered, where each observation has $P$ variables. We wish to model the data using a mixture of densities:

$$p(X_n|\theta = \{\theta_1, \dots, \theta_K\}, \pi) = \sum_{k=1}^{K} \pi_k f(X_n|\theta_k) \tag{1}$$

independently for each $n = 1, \dots, N$. Here, $f(\cdot)$ is a probability density function such as the Gaussian density function or the categorical density function, and each component has its own weight, $\pi_k$, and set of parameters, $\theta_k$. The component weights are restricted to the unit simplex, i.e., $\sum_{k=1}^{K} \pi_k = 1$. To capture the discrete structure in the data, we introduce an allocation variable, $c = [c_1, \dots, c_N]^\top$, $c_n \in [1, K] \subset \mathbb{N}$, to indicate which component a sample was drawn from, introducing conditional independence between the components,

$$p(c_n = k) = \pi_k, \tag{2}$$
$$p(X_n|c_n = k, \theta_k) = f(X_n|\theta_k). \tag{3}$$

The joint model can then be written

$$p(X, c, K, \pi, \theta) = p(X|c, \pi, K, \theta)p(\theta|c, \pi, K)p(c|\pi, K)p(\pi|K)p(K). \tag{4}$$

We assume conditional independence between certain parameters such that the model reduces to

$$p(X, c, \theta, \pi, K) = p(\pi|K)p(\theta|K)p(K) \prod_{n=1}^{N} p(X_n|c_n, \theta_{c_n})p(c_n|\pi). \tag{5}$$

In terms of the hierarchical model this is:

$$X_n|c_n, \theta \sim F(\theta_{c_n}), \tag{6}$$
$$c_n|\pi \sim \text{Categorical}(\pi_1, \dots, \pi_K), \tag{7}$$
$$\pi_1, \dots, \pi_K \sim \text{Dirichlet}(\alpha/K, \dots, \alpha/K), \tag{8}$$
$$\theta_k \sim G^{(0)}, \tag{9}$$

where $F$ is the appropriate distribution (e.g., Gaussian, Categorical, etc.) and $G^{(0)}$ is some prior over the component parameters. In practice, $\theta_k$ represents the parameters characterizing each cellular compartment (e.g., mean protein abundance profiles), $\pi$ captures the relative sizes of compartments, and $c_n$ indicates which compartment each protein belongs to. The hierarchical structure allows uncertainty to propagate through all levels of the model. We use the empirical priors suggested by Fraley and Raftery [43] for the Gaussian mixture models. For the Categorical model we use the average proportion of each class for each feature to define our prior density.

**T-augmented Gaussian mixture model.** Crook et al. [23] proposed capturing outliers within the Gaussian mixture model using a heavy-tailed multivariate t (**MVT**) distribution in their t-augmented Gaussian mixture (**TAGM**) model. They consider each component a mixture of a multivariate normal (**MVN**) density and this MVT which has common parameters across all components, i.e.,

$$p(X_n|c_n = k, \cdot) = \begin{cases} f_t(X_n|M, V, \eta), & \text{if the } n^{th} \text{ item is an outlier} \\ f_{\mathcal{N}}(X_n|\mu_k, \Sigma_k) & \text{otherwise,} \end{cases} \tag{10}$$

where $\mu_k$ and $\Sigma_k$ are the mean vector and covariance matrix of the $k^{th}$ component, and $M, V$ and $\eta$ are the parameters of the MVT which are set empirically and not sampled. Being considered an outlier is an inferred quantity indicated through the $\psi = [\psi_1, \dots, psi_N]^\top$ binary variable. Note that the items of known label (in our applications, the marker proteins) are never considered outliers. This extends the hierarchical model described above to:

$$(X_n|c_n = k, \psi_n\mu_k, \Sigma_k) \sim N(\mu_k, \Sigma_k)^{\psi_n} \mathcal{T}(\eta, M, V)^{1-\psi_n}, \tag{11}$$
$$c_n|\pi \sim \text{Categorical}(\pi_1, \dots, \pi_K), \tag{12}$$
$$\psi_n|\epsilon \sim \text{Bernoulli}(1 - \epsilon), \tag{13}$$
$$\pi_1, \dots, \pi_K \sim \text{Dirichlet}(\alpha/K, \dots, \alpha/K), \tag{14}$$
$$(\mu_k, \Sigma_k) \sim \mathcal{NIW}(\mu_0, \kappa, \nu, \Psi), \tag{15}$$

where we set $\eta$ to 4 to ensure heavy-tails, and set $M$ to empirical mean and $V$ to half of the global variance of the data following Crook et al. [23].

**Gaussian process.** The Gaussian Process (**GP**) is a generalisation of the Gaussian probability distribution over functions or infinite vectors rather than finite vectors. The GP is immensely flexible and can describe measurements of arbitrary proximity, most relevant in modelling temporal or spatial data. Specifically, the GP is a continuous stochastic process with index set $T$ such that its realisation at any finite collection of points $\{t_1, \dots, t_P\} \subset T$ is jointly Gaussian. It is uniquely

specified by a mean function $m(t)$ and a positive semi-definite covariance kernel $C(\cdot, \cdot)$, which determine the mean vectors and covariance matrices of the associated multivariate Gaussian distributions. That is, $f(x)$ is a GP with mean function $m(t)$ and covariance kernel $C(\cdot, \cdot)$ if for any $\{t_1, \ldots, t_P\} \subset T$

$$f(t_1, \ldots, t_P) \sim \mathcal{N}((m(t_1), \ldots, m(t_P)), \Sigma), \tag{16}$$

$$\Sigma_{ij} = C(t_i, t_j). \tag{17}$$

GP models have been used in a range of problems such as systems biology analyses [44–48] environmental and health studies [49,50], and monitoring patient health [51,52] and are built upon rigorous theoretical foundations in terms of support and contraction rates [53–55].

In practice, a prior mean function of $m(x) = 0$ is commonly used and the covariance function is chosen from a small family of easy to implement functions such as the squared exponential function [56]. Lacking any strong belief about periodicity, symmetry or other constraints on the functions, we use the squared exponential function which is defined for two time points/locations $t, t'$, as

$$c(t, t') = a^2 \exp\left\{-\frac{d(t, t')^2}{2l^2}\right\}, \tag{18}$$

where $a$ and $l$ are the amplitude and length-scale respectively. We use the most common choice for the metric $d(\cdot, \cdot)$, Euclidean distance or the $l^2$-norm.

**Gaussian process mixture models.** If we consider the mean vector of a Gaussian density as the finite realisation of an infinite function in our feature space, we can place a Gaussian process prior over the space of functions it is drawn from. Thus, for the $k^{th}$ component of this mixture, mean vector $\mu_k$ and observed data which has been transformed to a vector, $X_k = [X_{k_1}, \ldots, X_{k_{N_k}}]$,

$$X_k | \mu_k, \sigma_k \sim \mathcal{N}(\mu_k, \sigma_k^2 I_P), \tag{19}$$

$$\mu_k | a_k, l_k \sim GP(0, C_k). \tag{20}$$

As we assume exchangeable data for ease of notation we permute the original dataset such that we can denote $X_k$ with contiguous indices, i.e., $X_k = [X_1, \ldots, X_{N_k}]$. The posterior distribution can be derived by using the properties of the multivariate Gaussian relating to conditional distribution of two jointly distributed Gaussian variables [57],

$$\begin{bmatrix} y \\ f(X^*) \end{bmatrix} \sim \mathcal{N}\left(\mathbf{0}, \begin{bmatrix} C(X, X) + \sigma^2 \mathbf{I} & C(X, X^*) \\ C(X, X^*) & C(X^*, X^*) \end{bmatrix}\right), \tag{21}$$

$$\implies \begin{cases} \mathbb{E}(f(X^*)|y) & = C(X^*, X)(C(X, X) + \sigma^2 \mathbf{I})^{-1} y, \\ \mathbb{C}ov(f(X^*)|y) & = C(X^*, X^*) - C(X^*, X)(C(X, X) + \sigma^2 \mathbf{I})^{-1} C(X, X^*). \end{cases} \tag{22}$$

We can then say

$$(\mu_k | X_k, \sigma_k, a_k, l_k) \sim GP(m_k, \tilde{C}_k), \tag{23}$$

$$m_k = C_k[1:P](C_k + \sigma_k^2 I_{N_k P})^{-1} X_k, \tag{24}$$

$$\tilde{C}_k = C_k[1:P, 1:P] - C_k[1:P,](C_k + \sigma_k^2 I_{N_k P})^{-1} C_k[, 1:P]. \tag{25}$$

This involves an expensive calculation, taking the inverse of an $N_k P \times N_k P$ matrix. There are methods to scale GP models, e.g., sparse approximations [58] and low-dimensional approximations [59], see Liu et al. [60] for a review of the subject. However, in our application we have rich structure in the data as all items within a modality have the same time/spatial measurements with consistent distance between each measurement and we can use this to reduce the computational load without using any of these more complex methods [61], we can instead rewrite the posterior parameters as

$$m_k = \frac{N_k}{\sigma_k^2}(A_k - A_k \times (I_p - Q^{-1}))\bar{X}_k, \tag{26}$$

$$\tilde{C}_k = A_k - \frac{N_k}{\sigma_k^2}(A_k - A_k \times (I_p - Q^{-1}))A_k. \tag{27}$$

Please see section A.1 in S1 Text for this derivation.

We sample the hyper-parameters $(a^2, l^2, \sigma^2)$ according to a Metropolis-Hastings scheme. Standard log-normal hyperpriors were used.

**Multiple dataset integration.** MDI is a Bayesian integrative or multi-modal clustering method. Bayesian statistics allows quantification of uncertainty through the use of probability distributions and offers a modular framework for data analysis by making dependencies between data and parameters explicit [62]. In MDI, signal sharing is defined by the prior on the cluster label of the $n^{th}$ observation in the $M$ modalities/datasets:

$$p(c_n^{(1)}, \ldots, c_n^{(M)}|\cdot) = \prod_{m=1}^{M} \gamma_{c_n^{(m)}}^{(m)} \prod_{m=1}^{M-1} \prod_{l=m+1}^{M} (1 + \phi_{(m,l)} \mathbb{I}(c_n^{(m)} = c_n^{(l)})), \tag{28}$$

where $c_n^{(m)}$ is the label of the $n^{th}$ observation in the $m^{th}$ modality, $\gamma_k^{(m)}$ is the weight of the $k^{th}$ cluster in the $m^{th}$ modality, $\phi_{(m,l)}$ is the similarity between the clusterings of the $m^{th}$ and $l^{th}$ modalities and $\mathbb{I}(x)$ is the indicator function returning 1 if $x$ is true and 0 otherwise. Attractively, $\phi_{(m,l)}$ is inferred from the data and if there is no shared signal it will tend towards 0 giving us no dependency between these modalities in which case each dataset will have independent clusters.

Within each dataset/modality MDI uses a specific density to model the probability of allocation to a specific cluster. For example, one might wish to model continuous data with Gaussian densities, categorical data with Categorical distributions and/or time-series data with Gaussian processes. Thus we have flexibility in how we model each dataset within the MDI framework.

We implement this in C++ with the novel functionality to allow for observed classes and labels in a subset of datasets/modes and make this available through the R package `MDIr`. In the Bayesian paradigm, moving from an unsupervised cluster method to a semi-supervised method can be phrased as a missing data problem. Let $I = \{i_1, \ldots, i_N\}$ be a vector indicating if the label $c_n$ is observed or not. If $i_n = 0$ (i.e. the label is unobserved), then $c_n$ is sampled as usual, but if $i_n = 1$ then $c_n$ is known and held fixed. Observing labels means that we can no longer arbitrarily switch labels as the labels are inherently linked to a specific class (in our case, subcellular niches). The Gibbs sampler to perform inference on the model is implemented in C++ and includes a correction to the conditional distribution of the $\phi(m, l)$ parameters which was incorrect in previous implementations as observed by Stephen Johnson, Daniel Henderson, and Richard Boys in 2017 (private correspondence; see section A.2 of S1 Text).

As per Kirk et al. [40], we use an overfitted mixture model [63] to approximate a Dirichlet process mixture model [64–67] in each unsupervised modality, allowing inference of the number of clusters/classes present. If a number of classes are observed but one believes that additional classes might be present in the data one may use a semi-supervised overfitted mixture to detect novel classes as per Crook et al. [68].

**Markov chain Monte Carlo methods.** We use Markov chain Monte Carlo (**MCMC**) methods [69–72] to perform inference on our Bayesian models. MCMC is the gold standard for Computational Bayes [73–76], constructing a Markov chain of sampled parameters to describe the conditional posterior density for model parameters, with asymptotic guarantees that this description is exact. However, in practice MCMC is prone to becoming trapped in local modes of the target density. Designing algorithms that perform better in this scenario is an area of active research [77–83], but implementing these methods can be time-consuming and with no guarantee of overcoming the problem. We instead use Gibbs sampling [84] in our implementation of MDI and the consensus clustering approach to circumvent multi-modality [85]. We can use the collection of sampled values to infer a point estimate and provide a quantification of uncertainty about this. Common choices of point estimate for continuous parameters are the median or the mean, but to obtain a summary clustering we instead use that which minimises the variation of information [86], which has been found to have desirable properties [87], though alternative approaches exist [88,89]. To summarise the uncertainty about a partition, the posterior similarity matrix (**PSM**) is often used. This considers the average co-clustering probability of each pair of items based on the sampled clusterings. It is an $N \times N$ symmetric matrix. For a classification, we assign the samples to the class with the highest average probability of allocation across the retained MCMC samples.

**Transfer learning using a *k*-nearest neighbours framework.** The $k$-nearest neighbours (**KNN**) classifier is a supervised method in which predictions are made using the majority vote of an item's $k$ nearest labelled neighbours. The KNN-transfer learning (**KNN-TL**) classifier extends this to transfer information from an auxiliary modality to a primary modality [41,90]. To use the transfer learning algorithm, the best choice of $k_1, k_2$, the number of neighbours considered when classifying items with missing labels for each modality, must be estimated from the KNN classifier to be used as inputs into the KNN-TL classifier. This is performed via a grid search on a pre-specified grid, with the optimal value selected through cross-validation. Breckels et al. [41] adapted this method to predict protein localisation using LOPIT data as the primary dataset and GO terms as the auxiliary dataset (in contrast to MDI, the KNN-TL classifier does consider an ordering of datasets). In this context, this algorithm finds the $k_1$ nearest marker proteins to the $n^{th}$ protein in the LOPIT data and the $k_2$ nearest marker proteins in the GO data. The representation of each class among these neighbours is calculated, i.e., if $Z_1, Z_2$ are the sets of neighbours in the first and second modalities respectively,

$$p(c_n^{(v)} = j) = \frac{1}{|Z_v|} \sum_{i \in Z_v} c_i^{(v)} = j, \text{for } j = 1, \dots, K^{(v)}, v = 1, 2, \tag{29}$$

from which a localisation is predicted

$$p(c_n = j) = \theta_j p(c_n^{(1)} = j) + (1 - \theta_j) p(c_n^{(2)} = j) \tag{30}$$

where $\theta \in [0, 1]$ is estimated using a grid search (again, the optimal value is selected through cross-validation on the marker proteins). The final estimate is $\hat{c}_n = \arg\max_j p(c_n = j)$.

## Materials

**Simulation study.** Our simulation study investigates the benefit of having observed labels available in a modality (such as marker proteins in a hyperLOPIT experiment), and the advantage of performing an integrative analysis rather than independent analyses of each modality. To achieve this, we construct three scenarios. Each scenario is defined by the density used to generate the clusters (the generative model) and by the $\phi$ vector which determines the similarity of the clustering structure across modalities. We first generate the labels indicating the cluster an item is generated from in each modality and then sample measurements based on this cluster's parameters. In every simulation, the first dataset is generated from six clusters, the second from seven clusters and the third from eight clusters. We generate two hundred labels

and data points in each datasets from this model:

$$p(c_n^{(v)} = k | \gamma, \phi) \propto \gamma_k \prod_{w=1; w \neq v}^{V} (1 + \phi_{v,w}), \quad (31)$$

$$(X_n^{(v)} | c_n^{(v)} = k) \sim f(\theta_k). \quad (32)$$

There are fifteen measurements/feature for each data point (i.e., $N = 200, P = 15$ in each dataset in all simulations). For the semi-supervised models, a random 30% of the labels are given as observed in the first dataset with no labels or classes observed in the second and third datasets in all models. Within a generating seed the labels in each modality are the same across all scenarios and each dataset is informative of the others ($\phi = (12, 8, 4)$). The Adjusted Rand Index (**ARI**) [91] between the true labels of each pair of datasets across simulations is shown in Fig 1A. The ARI is a score comparing two partitions, adjusted for chance. A value of 0 means two partitions are no more similar than a pair of random partitions would be expected to be, 1 means they are identical.

We consider three different generating models which give the scenarios their names. The first two are

$$\text{Scenario 1 (Gaussian): } (X_n^{(v)} | c_n^{(v)} = k) \sim \mathcal{N}(\mu_k, \Sigma_k), \quad (33)$$

$$\text{Scenario 2 (MVT): } (X_n^{(v)} | c_n^{(v)} = k) \sim t_{\eta_k}(\mu_k, \Sigma_k). \quad (34)$$

Scenario 3, the Log-Poisson case, is more complex.

$$(Y_{n,p} | c_n^{(v)} = k) \sim Poisson(\lambda_{k,p}), \quad (35)$$

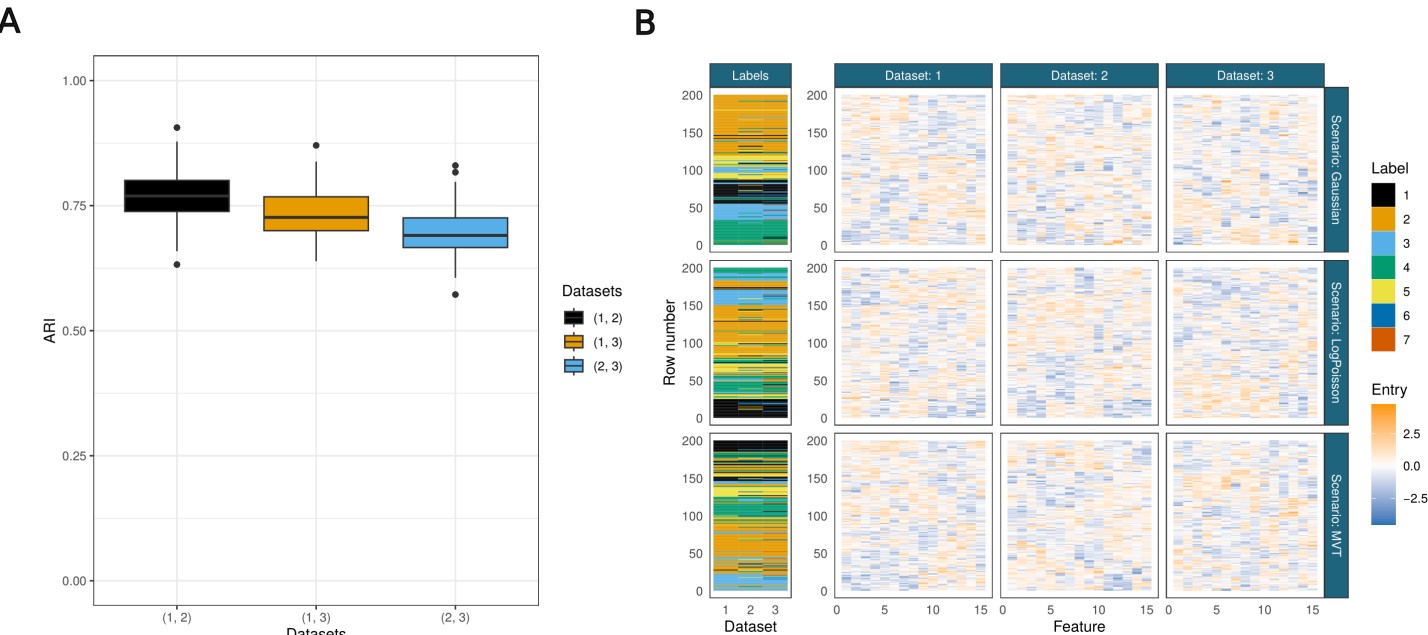

**Fig 1**. **A. ARI between the generating labels for each combination of dataset pairings within a given simulation across all scenarios.** For example, a point contributing to the (1, 2) boxplot is the ARI between the generating clusterings for the first and second datasets for a given random seed and scenario. B. Example of simulated data from each scenario and the generating labels all ordered by the data in the first modality for each scenario.

$$\epsilon_{n,p} \sim \mathcal{N}(0, 1), \tag{36}$$

$$X_{n,p} = \log\left(1 + Y_{n,p}\right) + \epsilon_{n,p}. \tag{37}$$

This is inspired by the simulation study of Chandra et al. [92], but the Gaussian noise is added after the log-transform to ensure this transform is always possible. Each cluster strongly deviates from Gaussian in this case. An example of the simulated data is shown in Fig 1B. These distributions were selected to test robustness under different model assumptions: Gaussian represents ideal conditions matching model assumptions (often assumed for gene co-expression data), MVT tests heavy-tailed outlier scenarios common in proteomics, and Log-Poisson evaluates performance under severe distributional misspecification (based on log-normalization of count data such as RNA-seq). A full description of the generating mechanisms and choice of parameters to differentiate the datasets are given in section B of S1 Text.

We fit five different models in each simulation. These are distinguished with the information available to them in the first dataset and whether they jointly model the datasets (i.e., MDI) or model each dataset separately using a mixture model, and if the model has access to class labels and the true number of classes present (e.g., when the number of subcellular niches is unknown). This comparison allows us to isolate the benefits of: (1) integrative vs. independent modeling, (2) supervised vs. unsupervised learning, and (3) known vs. inferred cluster numbers. In all cases datasets 2 and 3 are unsupervised and the number of clusters present is inferred. For the overfitted approaches, we use the overfitted mixture model framework of [63], which includes more mixture components than expected and allows the data to determine the effective number of clusters through Bayesian model selection. This creates two variants of semi-supervised MDI: (1) semi-supervised MDI where the true number of subcellular compartments is provided as prior knowledge, and (2) overfitted semi-supervised MDI where this number must be inferred from the data alone. The overfitted approach is often relevant to real applications where the total number of subcellular niches may be unknown, while the standard approach provides an upper bound on performance when complete prior knowledge is available.

- Unsupervised MDI: MDI with no observed labels and the number of clusters unknown,
- Overfitted semi-supervised MDI: MDI with known labels, but the true number of clusters is hidden,
- Semi-supervised MDI: MDI with both observed labels and the true number of clusters known,
- Overfitted semi-supervised mixture model: mixture model with known labels, but the true number of clusters is hidden,
- Semi-supervised mixture model: mixture model with both observed labels and the true number of clusters known,

We run 10 chains of each method to each simulated dataset. Each chain is 15,000 iterations long with the first 7,500 discarded as burn-in. We thin to every 100th sample to reduce autocorrelation. The best chain for the MDI methods is determined by the chain with the highest mean ARI between the point estimate clustering and the truth across datasets whereas for the mixture model the chains are selected based on the highest ARI between the point estimate and the truth in each dataset (rather than using the same seed across datasets).

**Validation study.** We compare our approach to a state-of-the-art transfer learning method previously applied to LOPIT data, the KNN-TL classifier, using the marker proteins from four of the same datasets analysed by Breckels et al. [41]. This includes two datasets for *Arabidopsis thaliana*, the *Root Callus* [16] dataset (referred to hereafter as *Callus* to distinguish it from the second *A. thaliana* dataset) which contains 261 marker proteins and 16 fractions, and the *Root* dataset [93] which contains 185 marker proteins and six fractions. a dataset for human embryonic kidney fibroblast cells [94] which contains 404 marker proteins and 8 fractions, and a mouse pluripotent stem cell dataset [12] which contains 8 fractions for 387 marker proteins. These are available through the `pRolocdata` Bioconductor package. Fig 2 shows PCA plots of these proteins. These are paired with GO term data. GO terms record observed and inferred characteristics of gene and gene products in terms of three sub-ontologies. The cellular component (**CC**) the product localises to; its molecular function (**MF**), such as binding or catalysis; and the biological processes (**BP**) the product is involved,

e.g., chromatin binding. The data we use are the same as are available through the `pRolocdata` Bioconductor package [95]. This data was prepared by Breckels et al. [41] by using all possible GO CC terms associated to the proteins in the experiment to construct a binary matrix representing the presence/absence of a given term for each protein, for each experiment.

We model the proteomic datasets with a TAGM model and the GO terms using a categorical mixture model (see the supplementary material of [40] for a description of this model) within the MDI framework. We run five chains of the TAGM and MDI models for 15,000 iterations, discard the first 5,000 as warm-up, thin to every 50th and then combine samples across chains (i.e. construct an ensemble). We follow the vignette of Breckels et al. [94] for the KNN-TL algorithm. This involves identifying the inputs $k_1, k_2$ and $\theta$ in a grid search using cross-validation. We consider a range of $k_1, k_2 = (3, 5, 7, 9, 11, 13, 15, 17, 19)$ and $\theta_k = (0.0, 0.25, 0.50, 0.75, 1.0)$ for $k = 1, \dots K$. As all the combinations of $\theta$ must be considered this becomes computationally intensive as $K$ increases. We considered the smaller of either the full combination matrix or a random sample of 20,000 rows. We inspect performance across ten folds with approximately 30% of the marker proteins observed in each organelle and 70% of the localisations predicted.

We compare performance using the Brier loss, macro F1 score and accuracy (defined below). Accuracy is the number of predictions that match the ground truth divided by the total number of predictions; this is the simplest score we use as it does not account for either uncertainty or class size. The F1 score is a common measure of performance in binary classification examples that corrects for sensitivity and specificity. The variation considered here is an extension to the multi-class setting. Letting $\hat{c}_n$ denote the predicted class for the $n^{th}$ item and $c_n$ denote the true class, then

$$\text{Accuracy} = \frac{1}{N} \sum_{n=1}^{N} \mathbb{I}(\hat{c}_n = c_n), \tag{38}$$

$$\text{true positives (k)} = \sum_{n=1}^{N} \mathbb{I}(\hat{c}_n = k)\mathbb{I}(c_n = k), \tag{39}$$

$$\text{false positives (k)} = \sum_{n=1}^{N} \mathbb{I}(\hat{c}_n = k)\mathbb{I}(c_n \neq k), \tag{40}$$

$$\text{false negatives (k)} = \sum_{n=1}^{N} \mathbb{I}(\hat{c}_n \neq k)\mathbb{I}(c_n = k), \tag{41}$$

$$\text{F1 (k)} = \frac{2 \times \text{true positives}}{2 \times \text{true positives} + \text{false negatives} + \text{false positives}}, \tag{42}$$

$$\text{Macro F1} = \frac{1}{K} \sum_{k=1}^{K} \text{F1 (k)}, \tag{43}$$

$$\text{Brier loss} = \frac{1}{N} \sum_{n=1}^{N} \sum_{k=1}^{K} (\hat{p}(c_n = k) - z_{n,k})^2, \tag{44}$$

where $z_{n,k}$ is the $(n,k)^{th}$ entry in the one-hot-encoding of the true allocation matrix, and $\hat{p}(c_n = k)$ is the inferred probability of the $n^{th}$ item being allocated to the $k^{th}$ class. The KNN-TL classifier does not provide a measure of uncertainty and thus we use $\hat{p}(c_n = k) = 1$ for $\hat{c}_n = k$ and 0 otherwise. The Brier score captures how well the Bayesian methods quantify the uncertainty about the classification.

To explore performance on more modern datasets and show that MDI has application beyond analysing pairs of datasets (in contrast to the KNN-TL classifier) we also investigate performance in predicting allocation of the marker proteins in data from Beltran et al. [96] and Orre et al. [14]. The first is spatial proteomics data for human cells infected by Human Cytomegalovirus (**HCMV**) across 5 time points (24, 48, 72, 96 and 120 hours post infection) and contains 295

marker proteins with measurements in every time point. The second contains measurements for 3,330 marker proteins across five different human cancer cell lines (epidermoid carcinoma A431, glioblastoma U251, breast cancer MCF7, lung cancer NCI-H322, and lung cancer HCC-827). We use a TAGM model for each dataset within the MDI model and as in the previous datasets we consider performance over ten different splits of 30% localisations observed and the remaining 70% inferred.

***Toxoplasma gondii.*** The Apicomplexa phylum of highly successful obligate parasites that can invade practically all warm-blooded animals [97–101]. As such, the apicomplexans are the source of much scientific interest, particularly understanding the mechanisms by which they infect a host and evade the innate and adaptive immune responses. However, this phylum is deeply divergent from the canonical models of molecular biology [31]. This means that results and knowledge do not always translate well from these well-studied organisms to the Apicomplexa. *T. gondii* is a natural candidate for the model apicomplexan as it can be cultured continuously in a lab and is readily amenable to genomic engineering for targeted experiments [102]. Beyond implications for other apicomplexans such as *Plasmodium falciparum*, the cause of malaria, *T. gondii* is also of medical interest in and of itself being the cause of Toxoplasmosis which, in immunocompromised individuals and as a congenital disease in infected infants, can be life-threatening [103]. From a proteomic perspective, the apicomplexans are a fascinating group; they have a myriad of unique cell structures and compartments that play a key role in their success. For example, they contain an apical structure central to penetration and invasion of animal cells; this contains secretory compartments including the micronemes, rhoptries, and dense granules for the staged release of molecules required for successful invasion and establishment within the host cell [104–107]. Some of the de novo subcellular niches are formed during the invasion of the host cell by the parasite [108] and it is hoped that better understanding of these organelles and their formation during invasion might lead to novel, targeted treatments [109].

Motivated by the importance of these organisms and their complex cellular structure, we perform an multi-omic analysis of the model apicomplexan, *T. gondii*, combining hyperLOPIT data from Barylyuk et al. [31] and transcriptomic data for the cell-cycle over twelve hours of post-invasion parasite replication [110] (Fig 2). We use cell cycle time-series gene expression data due to recent evidence that tightly co-ordinated transcriptional programs are essential for proper organelle [111, 112], and we expect this to reveal reveal the transcriptional programmes of the genes involved in organelle formation and the coexpression patterns of the secreted proteins [113]. Additionally, our MDI framework does not force proteins with strong spatial proteomics evidence to relocalize based purely on temporal data - proteins most likely to benefit from integration are those with ambiguous spatial signals but strong temporal co-clustering with organelle-specific genes. The hyperLOPIT data consists of 3 experimental iterations, each containing 10 fractions. By jointly modelling gene expression data with hyperLOPIT data annotated using marker proteins we hope to draw on the temporal programmes of the cell that are specific to spatial niches, and in doing so further resolve protein functional cohorts. However, jointly modelling the two datasets might also shed light on the localisation of proteins. As it stands, proteomes of most parasite compartments remain poorly characterised. Even the locations of proteins of predicted function based on conserved sequences in other organisms are largely untested in apicomplexans, though recent work from Barylyuk et al. [31] has improved our knowledge enormously. To generate both datasets the parasites were grown in and purified from human foreskin fibroblasts. For the gene-expression dataset parasites were synchronised using thymidine following the protocol developed by Radke and White [114]. We consider only the genes which have products in both datasets, reducing to 3,643 genes modelled.

Within the MDI model we choose data-specific probability densities to capture the structure in each dataset. We model the hyperLOPIT data using a TAGM model, setting the number of components to 26 which is the number of subcellular niches with associated marker proteins. In the cell cycle data, we are interested in co-expression patterns. We therefore mean-centre and standardise the measurements for each gene rather than considering absolute transcript abundance. To capture the temporal nature of the data, this modality is modelled using a mixture of MVNs with a GP prior on the mean function, as described in section Gaussian process mixture models. The proposal window for the Metropolis-Hastings sampling of the Gaussian process hyperparameters was set by achieving an acceptance rate between 0.2 and 0.8 in a

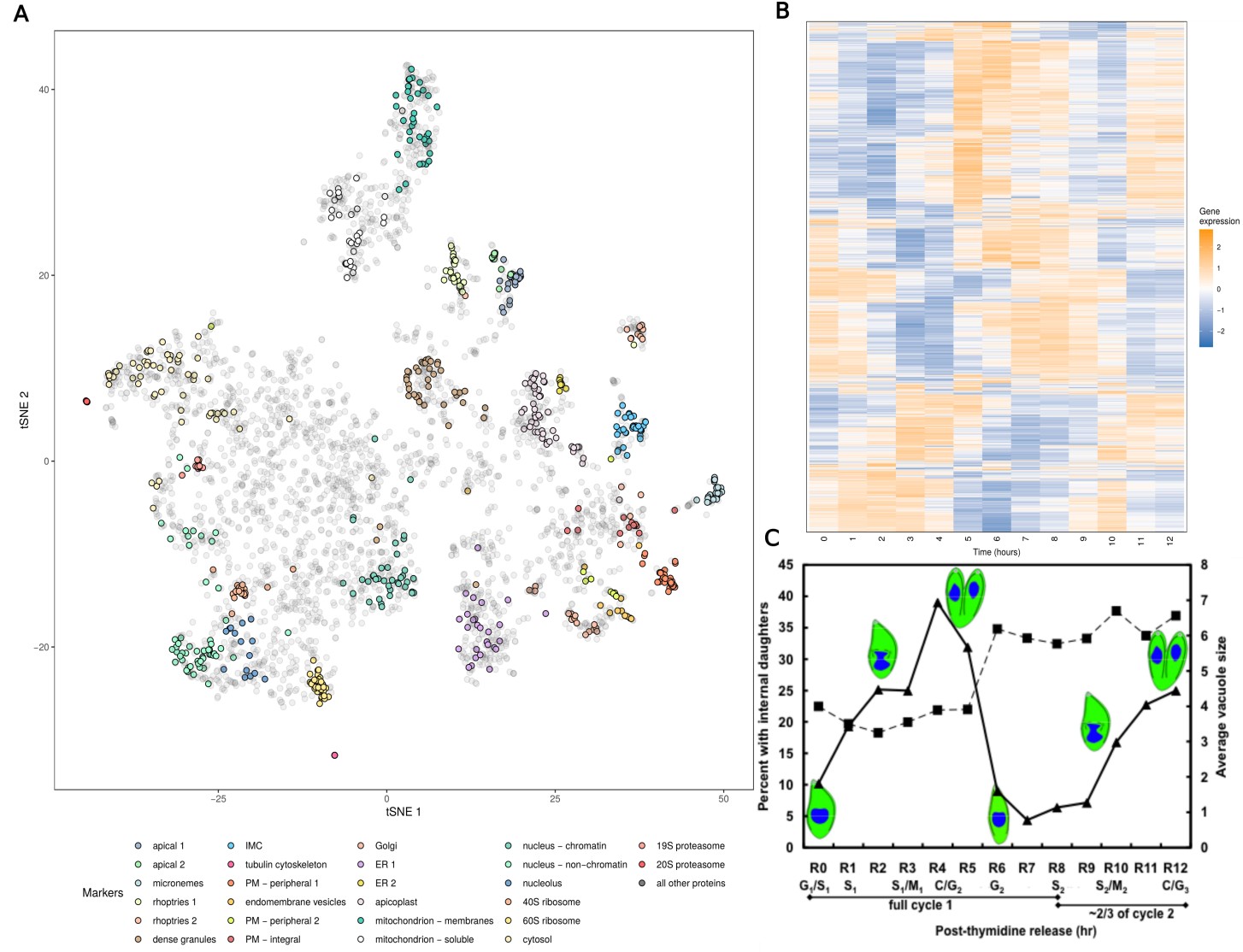

**Fig 2. The *T. gondii* data.** A) A tSNE of the hyperLOPIT data from Baryluk et al. with the marker proteins of each organelle highlighted. B) The gene expression data from Behnke et al. C) Fig 1 from Behnke et al. showing progression of cell number per vacuole (vacuole size, squares) and portion of cells undergoing division (triangles) at different timepoints in the gene expression data.

short chain. For the gene expression data we use 125 components to infer the number of clusters present (approximating a Dirichlet process mixture model [63] in this modality). We also model this data with an unsupervised mixture model with the same choices and parameters as the corresponding submodel in MDI, but the number of components required to be overfitted is much larger and we use 300 components in the mixture model. Due to the complexity of the data and a finite computational budget we used the consensus clustering approach of Coleman et al. [85] to circumvent the issue of poor mixing in individual chains and used the 15,000th sample from 150 randomly initialised chains (see section C in S1 Text for details of assessing stability). We inferred a point estimate partition for the hyperLOPIT data by using the localisation with the highest average allocation probability across MCMC samples, in the transcriptomic data we used the

estimate which minimised the lower bound to the posterior expected Variation of Information with a credible ball around the estimate as per the method of Wade and Ghahramani [86].

## Results

### Development of semi-supervised integrative models for spatial proteomics

To integrate a variety of datasets with spatial proteomics data we developed a semi-supervised Bayesian integrative model. This model allows the integration of a spatial proteomics specific Bayesian model [23,68] with Bayesian mixture models of other diverse data types. For example, categorical data can be integrated using a Categorical model and time-series data integrated using a Gaussian process based model. Our approach is flexible in that it allows labels/markers to be observed in none, all or one of the data-sets being integrated. Our model is built on the multiple dataset integration (MDI) framework [40], which means that information is shared using a parameter (which is learnt from the data) which up-weights the prior likelihood of two observations clustering together in each dataset. If the latent structures of two datasets are inferred to be unrelated, the model fits separate models in each. Our approach is built in the Bayesian framework and so quantifies uncertainty in parameters of interest. To enable others to apply our model to other datasets, it is implemented and disseminated as the R-package `mdir` (available from https://github.com/stcolema/MDIr).

### Simulation study

The predictive performance of the methods is shown in Fig 3. Semi-supervised MDI consistently outperforms all other approaches across datasets and scenarios, demonstrating the value of both integrative modeling and leveraging marker protein information. Notably, the overfitted semi-supervised MDI (which infers cluster number from data) performs as well as or better than the version with known cluster numbers, indicating robust automatic cluster detection under these simulation conditions. However, we note that in these examples the observed labels are sampled completely at random from the class, and that all classes are observed, enabling the overfitted mixture models to estimate $K$ accurately and that if this was not the case, e.g., if the sampling of the observed labels was biased in some way then performance might decline.

The benefits of integration are most pronounced under model misspecification. In the Log-Poisson scenario, where the Gaussian modeling assumptions are severely violated, semi-supervised MDI maintains superior performance while mixture models deteriorate. This robustness stems from two factors: (1) the $\phi$ parameter in MDI reduces reliance on distributional assumptions by upweighting cluster assignments based on cross-dataset consistency, and (2) semi-supervised learning anchors predictions using marker proteins rather than relying solely on distributional fit.

Unsupervised MDI performs surprisingly well, often matching semi-supervised mixture models even without access to marker proteins (similar median performance across scenarios). This highlights the power of leveraging shared structure across datasets—information from complementary modalities can compensate for lack of labeled examples. However, when both integration and supervision are available (semi-supervised MDI), performance is consistently optimal across all scenarios.

### Validation study

Having demonstrated that our semi-supervised model improves over other integrative mixture models, we wish to evaluate the performance of our method in spatial proteomics applications. We compare our integrative approach with TAGM - a non-integrative spatial proteomics model that can only see the spatial proteomics data. We also compare to the TL KNN method previously developed for integrative spatial proteomics. In this situation, we integrate the GO annotations, which are matrices of presence/absence of a particular annotation, with the MS-based spatial proteomics data. This constitutes a supervised task in both datasets.

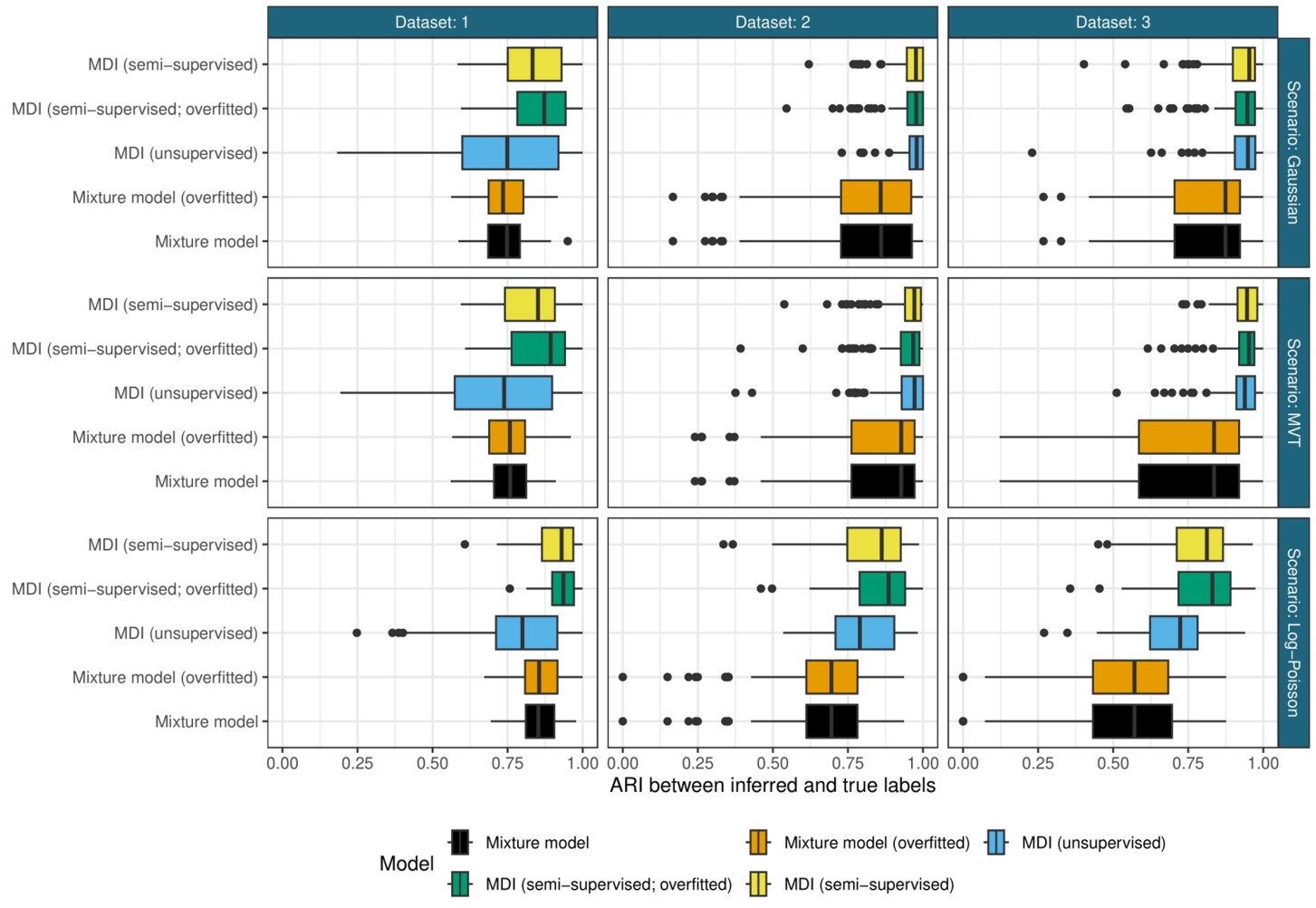

**Fig 3**. **Predictive performance of methods in the simulation study.** Semi-supervised MDI consistently outperforms other approaches across all scenarios, with overfitted semi-supervised MDI (which infers cluster number from data) performing as well as the standard version (with known cluster number). Horizontal facets are the different datasets, vertical facets are the different generative scenarios (Gaussian: ideal conditions; MVT: heavy-tailed data; Log-Poisson: severe model misspecification). The y-axis shows the different methods compared and the x-axis is the Adjusted Rand Index (ARI) between the inferred labels and the ground truth, where higher values indicate better performance. For fair comparison in the first modality the ARI is calculated on the same set of test proteins across all methods, excluding the marker proteins used for training in semi-supervised approaches. Each boxplot summarizes 100 simulation replicates.

In Fig 4 it can be seen that the integrative methods always match or outperform the TAGM model for all scores in all datasets for every test/train split, which is in keeping with the results from the simulation study. This is as expected for the informative second modality available to MDI and the KNN-TL algorithm in this study. MDI matches or outperforms the KNN-TL algorithm under all scores for the Human, Mouse and Callus datasets for all fractions of test/train splits. In the Root dataset the KNN-TL appears to perform slightly better than MDI under accuracy and the F1 scores, but they are more similar under the Brier loss, suggesting that the proteins that MDI mis-classifies have a high uncertainty for their localisation in this dataset. The uncertainty quantification and sampled distributions mean MDI and TAGM give much richer outputs than the KNN-TL algorithm. This demonstrates that our method is a significant advancement over previously developed approaches.

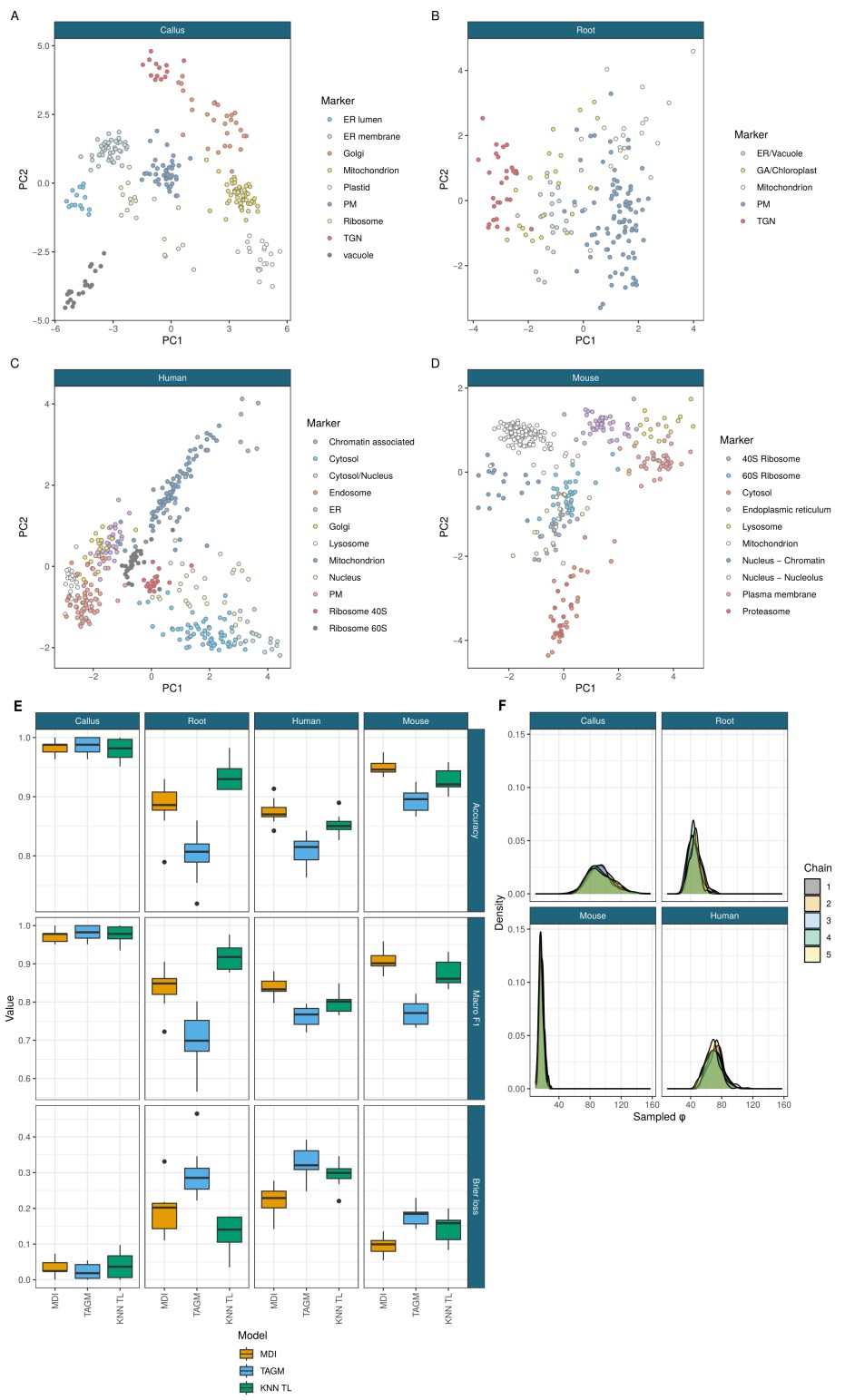

**Fig 4. A - D) PCA plots of the marker proteins from the different LOPIT datasets.** E) Performance of the MDI, TAGM and transfer learning models for the held out data across ten folds under accuracy, macro F1 and the Brier loss. F) The inferred integration parameter $\phi$ suggesting a high degree of information sharing.

To further assess the advantages an integrative approach, we examined two MS-based spatial proteomics studies with multiple experiments. The first was the spatio-temporal study of [96] in which human fibroblasts where infected with human cytomegalovirus (HCMV). Five spatial proteomics datasets were generated at timepoints of hours post infection (24, 48, 72, 96, 120). The focus of their study was to examine the differences between infection and control. Here, we integrate all five datasets in the infected experiment to demonstrate that predictive performance improves compared to considering each dataset in isolation compared with TAGM (see Fig 5). This demonstrates that our approach can also scale to many datasets, however the transfer KNN algorithm could not scale because of its combinatorial approach. As an additional example, [14] generated subcellbarcode - spatial proteomics datasets over five different human cancer cell lines. There approach used extensive offline chromatography and so the number of measured protein approaches close to 10,000 per cell line across 15 subcellular niches. We also integrated each of these cell lines and saw predictive performance increase in all of the datasets. This demonstrates that these atlases could be improved by borrowing information across related experiments to obtain more accurate predictions whilst allowing for difference between the datasets. In the next section, we demonstrate the increased flexibility of our approach by integrating a dataset without any annotations at all.

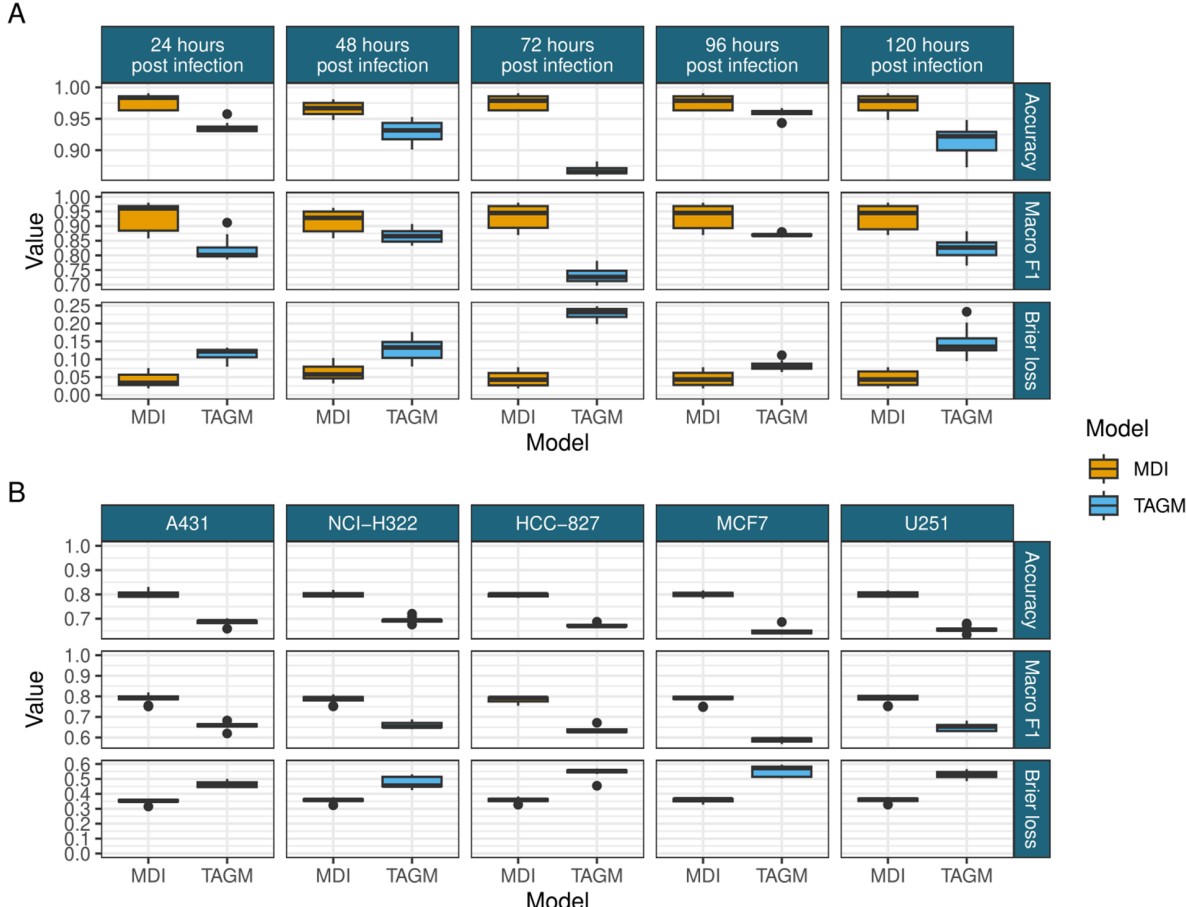

**Fig 5**. A) Comparison of the performance of the independent TAGM models and the MDI model in predicting a random test set of 70% of the marker proteins in each time point from Beltran et al. [96]. B) Comparison of the performance of the independent TAGM models and the MDI model in predicting a test set of 70% of the marker proteins in each cell line from Orre et al. [14].

## Integrating spatial proteomics and gene expression time-series data from *Toxoplasma gondii*

To demonstrate the versatility of our approach, we applied our semi-supervised integrative framework to *Toxoplasma gondii* spatial proteomics data and cell-cycle transcriptomics data. Briefly, the spatial proteomics is modelled using the TAGM model whilst the transcriptomic data us modelled using a mixture of Gaussian proccesses, these two models are then joined using the MDI framework. For additional modelling details, we refer to the method section.

The localisation predictions from this joint analysis, which included both spatial proteomics and time-series gene expression, are plotted in tSNE coordinates of the hyperLOPIT data visualised in Fig 6A. Broadly, the allocations continue to cluster in accordance with the tSNE coordinates demonstrating the strong localisation signal from the spatial proteomics data.

Subsequently, we focus on the behaviour of subset of proteins from secretory organelles and ER. For these subcellular niches the proteins that are allocated to them from our joint analysis are shown in Fig 6B. We would expect that the majority of the corresponding genes are "invasion-ready", highly expressed from beginning (invasion G/S-phase) of the cell-cycle and the invasion of the host's cell. However, the gene expression may not be high for some protein of the secretory organelles because they are already highly abundant. Thus, more specifically, gene-expression is high when the proteins need to be remade due to them being secreted or cellular division dilute their concentration and so more proteins are needed. This is the case for the proteins allocated to the rhoptries (a specialised organelle whose proteins are secreted during penetration) (Fig 6C). The proteins allocated to the micronemes and the apical complex compartments lag slightly behind that of the rhoptries. The lag in expression of micronemes tells us that rhoptries are made first, then new micronemes, and this displacement in time is interpreted to be because they use similar sorting routes through the Golgi and therefore the cell temporarily separates their synthesis so that they do not get mislocalised.

To assess the benefit of including the temporal data, we plot the allocation probabilities from the TAGM model applied to the spatial proteomics alone against the allocation probabilities from the integrative analysis for secretory organelles (Fig 6D). We observe a general trend where by the integrative analysis was more confident than the TAGM analysis with 73% of proteins having higher probabilities with the integrative approach. Only four proteins had different allocations between the two approaches but three of those where simply reallocated to the other apical cluster. TGME49_285150, a protein of unknown function, was reallocated to the apical rather than the dense granules albeit with low probability 0.64. A few proteins had much lower allocation probability with the integrative analysis including two proteins with a difference of more than 0.3. One of these was a protein of unknown function whilst the other was ERK7, which typically has an apical localisation but here allocated to the dense granules with low probability. ERK7 localisation appears to be AC9 dependent [115] and so the low probability here may be indicative of a diffuse subcellular localisation. These observations suggest that integrative analysis helps further calibrate our localisation analysis and inclusion of temporal datasets in future studies will produce a more refined and confident set of allocation.

We then explore the behaviour of the proteins predicted to localise to the dense granules, an important organelle in the apicomplexan post-invasion strategy for re-engineering the host cell environment to enable the parasites' safe growth and replication (Fig 7). These secretory vesicles are key in the formation of the parasitophorous vacuole in which the parasite resides. It was recently discovered that the related parasite *Cryptosporidium* boasts a small granule as a subset of the dense granules [116] and the question remains of whether in *Toxoplasma* dense granules also form a heterogeneous population of secretory compartments [117]. We consider the jointly inferred clusters of the transcripts corresponding to these proteins in the transcriptomic data, and find that the majority of these proteins are highly expressed towards the G/S transition stage of the cell cycle. However, there is heterogeneity in these dense granule gene expression programmes, for example a subset is more highly expressed closer to the S/M transition (Fig 7B, cluster 8) — more in line with the timing of biogenesis of small granules within *Cryptosporidium*. Our analysis suggests that there is additional heterogeneity within the dense granules and, depending on the scientific question, it might be beneficial to consider more homogeneous

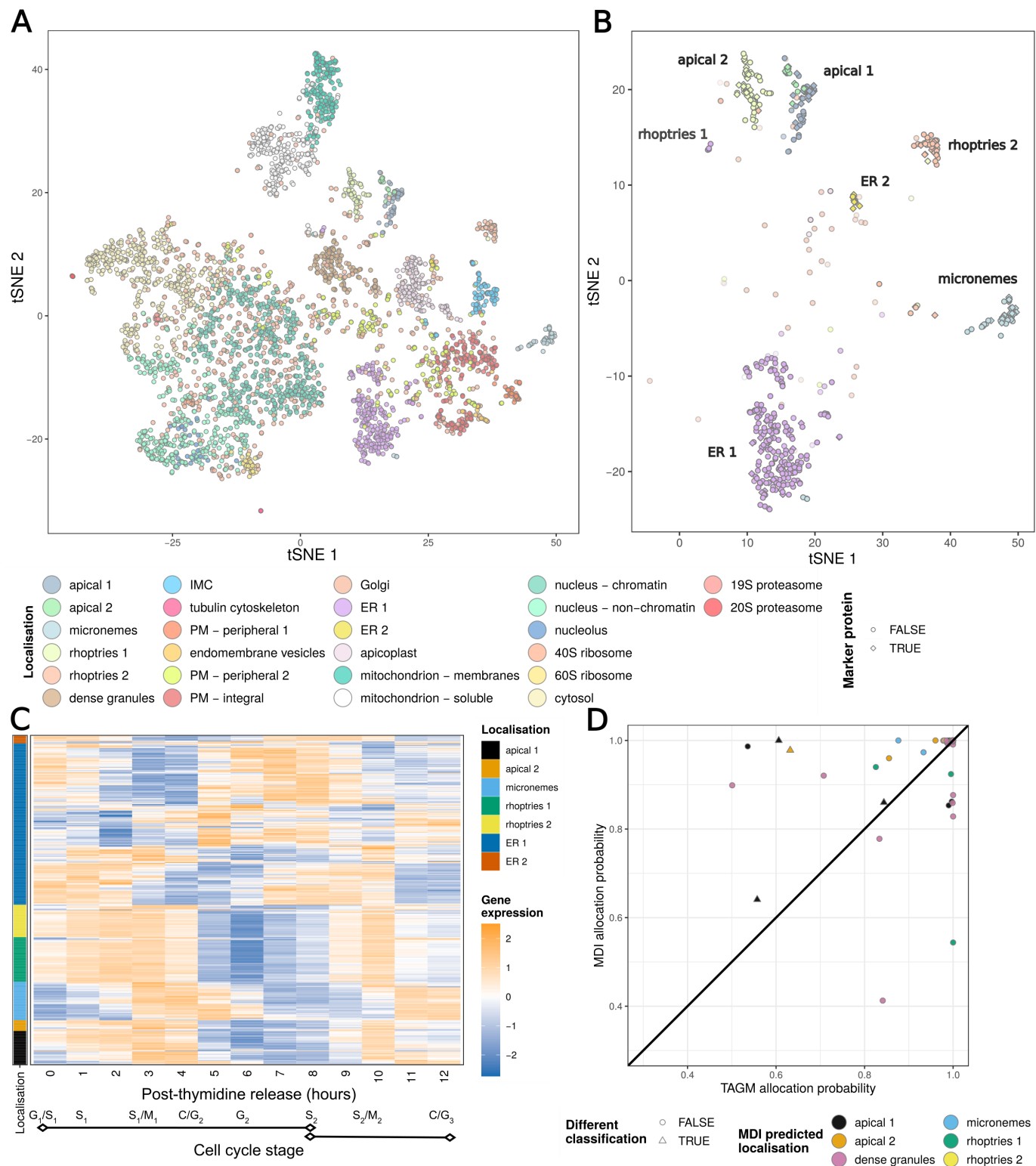

**Fig 6. A) tSNE projection of the hyperLOPIT data from Baryluk et al coloured by the predicted localisations using both the hyperLOPIT and gene-expression data.** B) The location in A) of the proteins predicted to localise to a subset of localisations (apical 1, apical 2, the micronemes, rhoptries 1, rhoptries 2, ER1 and ER2), opacity set by inferred probability of allocation. C) The gene co-expression data for these proteins annotated by the predicted localisation. D) Comparison of the predicted localisations and the associated probabilities from the MDI and TAGM models for these proteins.

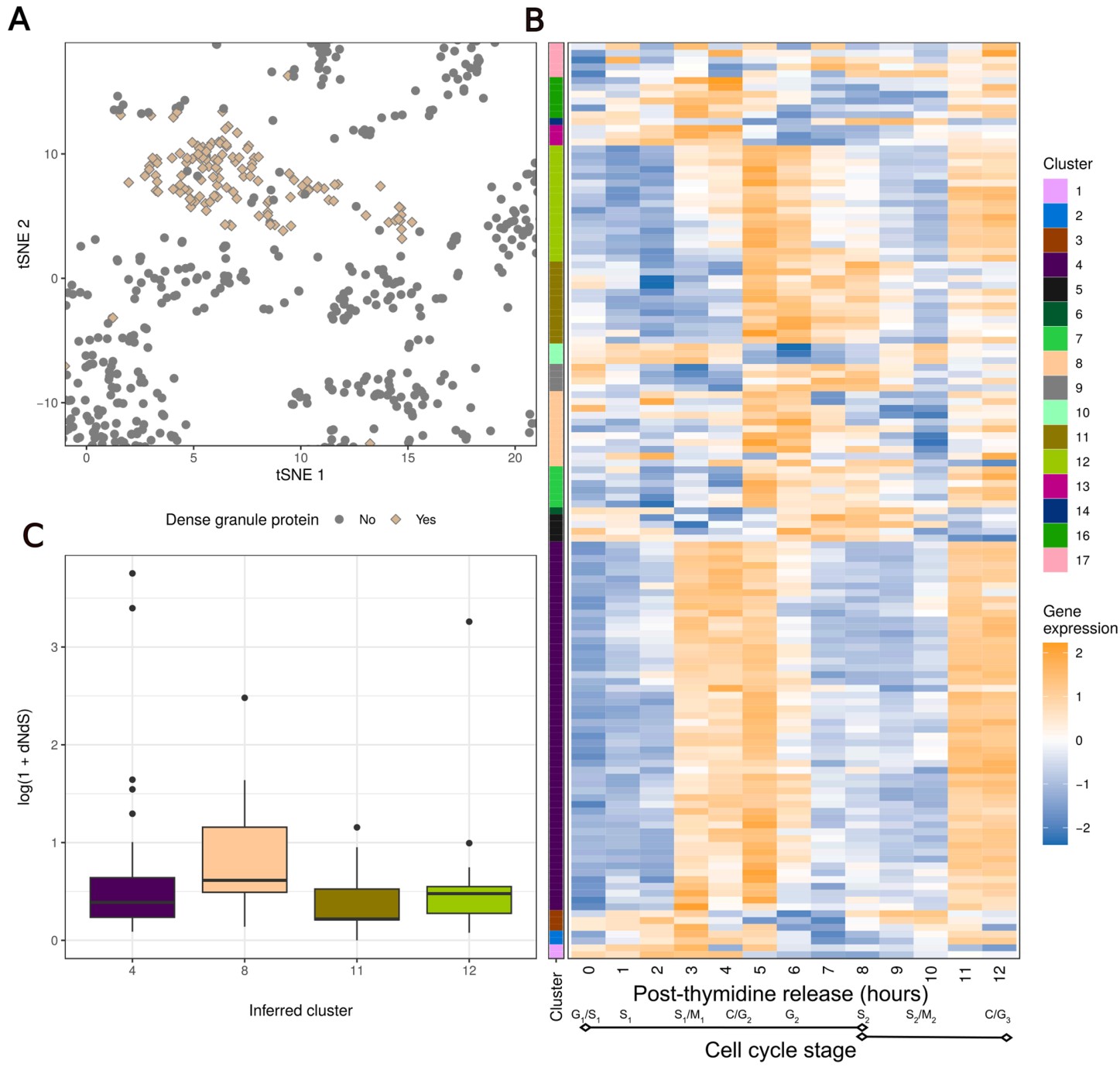

**Fig 7. A) The subsection of the tSNE plot for the hyperLOPIT data focusing on the proteins predicted to localise to the dense granules using both dataset.** B) The gene expression data for the genes whose members are predicted to localise to the Dense Granules, annotated by the point estimate clustering. The predicted localisation of the proteins in the hyperLOPIT data and the point estimate clustering in the gene expression data are jointly inferred with the same model run of the MDI model on these data. C) Comparison of the log ratio of dN to dS for the proteins from the 4 largest clusters from Fig B).

sub-niches within these heterogeneous bodies. Moreover, considering the distributions of the protein-average ratio of non-synonymous and synonymous mutation rates in the four largest clusters shows that cluster 8 has a significantly higher mean $d_N/d_S$ ratio than the other clusters (Fig 7C). This indicates that these proteins are under stronger selection pressures compared with the other dense granule proteins and, therefore, that there might be a distinctive role for this cohort of proteins. It is currently unknown if these differences might manifest only in their expression timing and levels, or if they might occupy distinctive spatial niches similar to the *Cryptosporidium* small granules from dense granules. But either way, our integrative analysis has uncovered a further layer of regulation, the first at the level of subcellular localisation and the second at the level of transcriptional regulation. These findings suggest that semi-supervised Bayesian integration can uncover previously unknown functional relationships in cell organisation.

## Discussion

We have introduced a semi-supervised, multi-modal Bayesian integrative method for spatial proteomics applications. We have shown on real and simulated datasets that it matches or outperforms state-of-the-art methods and offers a rich and informative output, particularly a principled quantification of uncertainty about the predictions. We then applied our method to a pair of 'omics datasets for *T. gondii*. Our analysis goes beyond that of Barylyuk et al. [31] where expression data is considered independently. We validated our analysis by considering the expression data for the secretory organelles, demonstrating that they correlated with their functions. We also found evidence suggesting that the dense granules may form a non-homogeneous functional population that have varying expression patterns during the cell cycle that are identified by transcriptional timing rather than on subcellular localisation alone.

Our approach is broadly applicable and various likelihood functions are implemented to allow integration of diverse data types, including discrete, continuous and time-series data with spatial proteomics data. We have shown that semi-supervised, integrative analysis is a useful tool for spatial proteomics and provided an R package as an open-source dissemination of our method. We envisage a broad range of applications of our method.

Our method has some limitations. Firstly, Bayesian methods are computationally intensive. Our method has complexity $O(DNKV)$ for $D$ MCMC samples, $N$ proteins, $K$ clusters, and $V$ datasets, plus $O(MKP^2)$ for updating covariance parameters. This becomes prohibitive for very large datasets, though users are provided with principled uncertainty quantification and the ability to integrate heterogeneous data types in return for this cost. Note that the consensus clustering approach we employ [85] significantly reduces computational burden by using many short chains rather than few long chains, avoiding the need for lengthy burn-in periods while capturing multiple-modes in the parameter space.

Secondly, whilst we have shown some robustness to model misspecification, our analysis is likely to be affected by gross misspecification of the likelihood. In the integrative case, the mis-specification of a model for a single dataset may have a negative impact on the modelling of all other datasets, as the misspecification "leaks" from one dataset into another where there is no misspecification, corrupting the analysis. In this case, one might apply "cut-models" or adapt our approach using an appropriate likelihood.

Third, prior specification requires careful consideration. We attempt to use uninformative or weak priors (e.g., for similarity parameters $\phi$) that allow the data to dominate inference, or empirical Bayes approaches (for component parameters) to encourage meaningful prior choices, but this does not remove the complexity and subjective choice of prior.

## Supporting information

**S1 Text. Semi-supervised Bayesian integration of multiple spatial proteomics datasets.** Model and computational derivations (Section A), simulation study descriptions (Section B), and additional model convergence analyses for the *Toxoplasma gondii* application (Section C). Includes supporting figures (A, B) and additional references.
(PDF)

## Author contributions

**Conceptualization:** Oliver M. Crook, Paul D. W. Kirk.

**Formal analysis:** Stephen Coleman, Oliver M. Crook.

**Funding acquisition:** Chris Wallace, Paul D. W. Kirk.

**Investigation:** Stephen Coleman.

**Methodology:** Stephen Coleman, Oliver M. Crook, Paul D. W. Kirk.

**Project administration:** Oliver M. Crook.

**Resources:** Kathryn S. Lilley.

**Software:** Stephen Coleman.

**Supervision:** Chris Wallace, Oliver M. Crook, Paul D. W. Kirk.

**Validation:** Stephen Coleman, Kathryn S. Lilley.

**Visualization:** Stephen Coleman.

**Writing – original draft:** Stephen Coleman, Lisa Breckels, Ross F Waller, Oliver M. Crook.

**Writing – review & editing:** Stephen Coleman, Lisa Breckels, Ross F Waller, Kathryn S. Lilley, Oliver M. Crook.

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
