## [Decision Letter · Decision Letter 0]

7 May 2025

PCOMPBIOL-D-25-00095

Semi-supervised Bayesian integration of multiple spatial proteomics datasets

PLOS Computational Biology

Dear Dr. Coleman,

Thank you for submitting your manuscript to PLOS Computational Biology. First I would like to apologise for the delay in the review process, this was largely a function of finding willing and knowledgeable reviewers. As you will see from the reports, both reviewers appreciated the value of your study and recommend only relatively minor edits to improve the clarity of manuscript, provide additional useful context and promote adoption of your method. Consequently we invite you to submit a revised version of the manuscript that addresses the excellent points the reviewers raised.

Please submit your revised manuscript within 30 days Jul 07 2025 11:59PM. If you will need more time than this to complete your revisions, please reply to this message or contact the journal office at ploscompbiol@plos.org. Please include the following items when submitting your revised manuscript:

We look forward to receiving your revised manuscript.

Kind regards,

John Parkinson

Guest Editor

PLOS Computational Biology

Mark Alber

Section Editor

PLOS Computational Biology

**Journal Requirements:**

At this stage, the following Authors/Authors require contributions: Stephen Coleman, Lisa Breckels, Ross F Waller, Kathryn S Lilley, Chris Wallace, Oliver M Crook, and Paul D.W. Kirk. Please ensure that the full contributions of each author are acknowledged in the "Add/Edit/Remove Authors" section of our submission form.

5) Thank you for stating "All data is available publicly from previous publications". Please provide a complete Data Availability Statement in the submission form, ensuring you include all necessary access information. If your research concerns data from external sources, please amend your Data Availability Statement to include the full links to the data.

1) Please clarify all sources of financial support for your study. List the grants, grant numbers, and organizations that funded your study, including funding received from your institution. Please note that suppliers of material support, including research materials, should be recognized in the Acknowledgements section rather than in the Financial Disclosure

2) State the initials, alongside each funding source, of each author to receive each grant. For example: "This work was supported by the National Institutes of Health (####### to AM; ###### to CJ) and the National Science Foundation (###### to AM)."

3) State what role the funders took in the study. If the funders had no role in your study, please state: "The funders had no role in study design, data collection and analysis, decision to publish, or preparation of the manuscript."

4) If any authors received a salary from any of your funders, please state which authors and which funders.

7) Your current Financial Disclosure states, "The author(s) received no specific funding for this work."

However, your funding information on the submission form indicates receiving funds. Please ensure that the funders and grant numbers match between the Financial Disclosure field and the Funding Information tab in your submission form. Note that the funders must be provided in the same order in both places as well.

**Reviewers' comments:**

Reviewer's Responses to Questions

Reviewer #1: The manuscript presents an innovative and well-supported approach to integrating spatial proteomics datasets using Bayesian models. The methodology is rigorous, and the results are compelling. Minor revisions to improve clarity, provide additional justification for certain methodological choices, and better contextualize findings in the broader field would strengthen the manuscript significantly.

Abstract

Line 20-21:

The statement "existing approaches... do not quantify uncertainty" is strong. Consider specifying which methods lack this feature to prevent overgeneralization.

Ensure "semi-supervised Bayesian approach" is briefly explained for a broader audience.

Line 24: What types of the data?

Introduction

Lines 78–88: Consider restructuring the paragraph that discusses multi-omic integration challenges to emphasize why Bayesian methods are particularly suited for this task.

A direct comparison between past integrative methods and the proposed approach would help contextualize the contribution.

Lines 95–97: The sentence “Our approach is applicable beyond MS-based spatial proteomics...” is strong but could be supported with an example outside proteomics.

Materials and Methods

The section introducing Gaussian mixture models (GMM) is mathematically rigorous but slightly dense.

Equations (6)–(9) are helpful, but it would be beneficial to provide a brief practical interpretation of each parameter.

lines 177–197

could include a brief comparison to other integration models or simpler methods into the introduction part.

Results

Simulation Study

The explanation of different generative models (Gaussian, MVT, Log-Poisson) is well-structured, but an intuitive justification for their selection (e.g., why these specific distributions?) would be helpful.

Figure 3 conveys important insights, but the differences between semi-supervised MDI and overfitted semi-supervised MDI should be better explained.

Discussion

The limitations section is somewhat brief. Potential areas to mention: Computational costs of methods, including MCMC for large datasets (big O complexity). Challenges in parameter selection and prior specification.

Reviewer #2: This manuscript by Coleman et al. addresses the problem of finding the subcellular localization of proteins in spatial proteomics experiments. Specifically, “LOPIT” type mass spectrometry experiments involves separating cellular proteins into sub-cellular fractions through multi-step centrifugation, followed by quantification of each fraction’s TMT profile using MS. Each organelle would have a different profile due to different density. The computational task then is to find the latent sub cellular localization from the TMT profile and potentially estimate uncertainty. This is typically done using supervised or semi-supervised classification approaches.

Here, the authors propose a multiple dataset integration framework that incorporates gaussian process modeling to incorporate different types of auxiliary data (categorical, time-series, etc.). This allows spatial proteomics mass spectrometry data to be combined with other data types (such as prior literature annotations) to improve classification performance. This work expands on a series of prior papers from the authors. Most notably, Crook et al. PLoS Comp Biol 2018 which introduced the Bayesian mixture model approach to analyze TMT fraction profiles, and also introduced the outlier T-distribution to catch non-conforming data (TAGM approach). Crook et al. PLoS Comp Biol 2020 then extended this approach toward semi-supervised discovery of new cluster/localization. Crook et al. Annals of Applied Statistics 2022 introduced the Gaussian Process mixture modeling approach. Breckels et al. PLoS Comp Biol 2016 introduced the KNN-TL classifier to include GO terms in compartment assignment.

Overall this looks to be an excellent paper that addresses an important need. As the authors stated, the sub cellular localization of proteins is a major determinant of their function. How to determine the dynamic localization of proteins in an unbiased and context-specific manner remains an incompletely solved problem. The proposed approach here appears to be robust and well justified. The authors demonstrate performance using both simulated data and existing experimental data sets. The rationale and notations are well explained, and the manuscript is well written. Another major strength is the availability of an R package so other investigators can take advantage of this advance.

I have no problem recommending acceptance, and only have the following minor comments. These are not necessary for acceptance but the authors’ considerations.

Comments:

1. The primary goal of this work, as I understand it, or at least as laid out in the abstract/introduction, is to improve on the inference of protein sub cellular localization. This appears to have evolved somewhat as the manuscript progresses to the T. gondii results section. It would be nice to have more details on why the authors opted to incorporate the time-series data as opposed to other types of data (e.g., co-expression or interactome data), and intuitively how this may help with localization assignment. When the protein assignment changes this way (e.g., ERK7 or the dense granule proteins), does it indicate a change in actual sub cellular localization, or that different proteins within the same localization can take on different function or temporal behavior?

2. Since the AOAS 2022 paper already described the GP model in some detail, it would be nice to have more clarity on how the current method compares, and also include the semi-supervised GP model as comparison along with the TAGM mixture models in benchmarking (e.g., in Figures 3-5). Also as far as I know, previous methods from the authors already had the capacity to integrate multiple independent sets of LOPIT TMT data. To show the value of multi datatype integration, it would be nice to directly compare whether incorporating

3. Did the authors investigate the effect of GO categorical annotation on difficult-to-localize outliers, e.g., if a protein is annotated to be in either the cytosol or the nucleus in GO or UniProt, would the performance gain in protein assignment reflect this by preferentially localizing a protein to those compartments? It would be interesting as well if the authors could comment on whether this integration can help the assignment of differential localization between the two annotated compartments.

4. I was not completely clear on how accuracy and other performance metrics were calculated in the validation study. Were the true positives used to calculate these metrics picked from holdout markers that were then not used to train the supervised/semi-supervised models? As far as I know training the TAGM mixture model requires 20-30% of proteins to be designated as markers and how the markers are chosen can have an effect on the classification outcome. Did the authors investigate whether the change in markers affected the TAGM mixture model and the MDI approach differently? (For instance, it is not clear how the use of GO annotations for integration affects marker selection. Is manual marker selection still required for the MDI method, or can literature annotation be used to skip manual markers?)

5. Can the authors explain in greater details what the phi parameter is and how it is estimated, and how 12, 8, 4 were chosen for the simulation study?

6. A major strength here is the implementation of an R package but the Github of MDIr appears to contain little to no documentation. While the repository is not strictly part of the paper, I believe this should be fixed to make the paper more useful to the readers. Besides documentations and/or tutorial, it would also be useful to have more discussion on how users may apply their own data - e.g., computational requirements, what type of mass spec data is allowed, how to tune parameters, etc.

7. It would be useful to have additional discussion on how robust the method is to data input (e.g., some existing LOPIT data are rather sparse with low mass spec depth, others may use suboptimal or misannotated markers, etc. etc.)

8. Some typos/formatting errors, e.g., line 37 time-course; supplement line 56 and 105, missing ref.

**Have the authors made all data and (if applicable) computational code underlying the findings in their manuscript fully available?**

Reviewer #1: Yes

Reviewer #2: Yes

PLOS authors have the option to publish the peer review history of their article (what does this mean?). If published, this will include your full peer review and any attached files.

Reviewer #1: No

Reviewer #2: No

**Figure resubmission:**
---

## [Decision Letter · Decision Letter 1]

30 Nov 2025

Dear Mr Coleman,

We are pleased to inform you that your manuscript 'Semi-supervised Bayesian integration of multiple spatial proteomics datasets' has been provisionally accepted for publication in PLOS Computational Biology.

Best regards,

Mark Alber, Ph.D.

Section Editor

PLOS Computational Biology

Mark Alber

Section Editor

PLOS Computational Biology

Reviewer's Responses to Questions

**Comments to the Authors:**

Reviewer #1: The authors have addressed my concerns

Reviewer #2: The authors have addressed my previous (minor) comments to my satisfaction. I applaud the authors for this excellent contribution.

**Have the authors made all data and (if applicable) computational code underlying the findings in their manuscript fully available?**

Reviewer #1: Yes

Reviewer #2: Yes

PLOS authors have the option to publish the peer review history of their article (what does this mean?). If published, this will include your full peer review and any attached files.

Reviewer #1: No

Reviewer #2: No

---

## [Editor Report · Acceptance letter]

PCOMPBIOL-D-25-00095R1

Semi-supervised Bayesian integration of multiple spatial proteomics datasets

Dear Dr Coleman,

I am pleased to inform you that your manuscript has been formally accepted for publication in PLOS Computational Biology. Your manuscript is now with our production department and you will be notified of the publication date in due course.

With kind regards,

Anita Estes
